# Differences in Visual Preference in Rural Landscapes on the Plain of La Mancha in Spain

**Esperanza Ayuga-Téllez** [1] , **Juan José Ramírez-Montoro** [2] **, Maria Ángeles Grande-Ortiz** [2,*] 
**and Diego Muñoz-Violero** [2]

1   Buildings, Infrastructures and Projects for Rural and Environmental Engineering (BIPREE),
    Universidad Politécnica de Madrid, 28040 Madrid, Spain; esperanza.ayuga@upm.es
2   Departamento de Ingeniería y Gestión Forestal y Ambiental, ETSI de Montes,
    Universidad Politécnica de Madrid, 28040 Madrid, Spain; juanjose.ramirez.montoro@upm.es (J.J.R.-M.);
    dimuvio@yahoo.es (D.M.-V.)
*   Correspondence: m.angeles.grande@upm.es

**Abstract:** For centuries, agricultural activities have marked and defined the landscape with its own distinctive features. The consideration of the rural landscape as a resource has gained traction in recent years. In Europe, the European Landscape Convention offers a solid framework that places landscape at the forefront of European policies on cultural heritage, environment, and territorial ordination. The most important new development is the integrated vision of the landscape in its cultural and natural aspects, and the introduction of its social dimension. This work analyses the influence of different factors on preferences for rural landscapes in the locality of Campo de Criptana (Ciudad Real), representative of the singular rural landscape of the La Mancha plain. The method for assessing landscape is the people's aesthetic response to it. Specifically, an analysis has been made of the observers' preferences in relation to their educational level (university educated or not), gender, age, and place of origin (whether they come from the locality itself or from outside). This is one of the few works that analyse the place of origin of the observer. In view of these results, it can be concluded that all the demographic factors analysed have an influence on preferences in rural landscapes.

**Keywords:** landscape quality; assessment; public preferences; demographic factors

## 1. Introduction

The concept of landscape is difficult to characterise, as it describes a multipurpose element [1] whose definition has evolved over time. Hull and Ravell [2] defined landscape as the external, natural, or anthropic environment that can be directly perceived by a person visiting and using that environment. The European Landscape Convention (ELC) [3] defines landscape as "an area, as perceived by people, whose character is the result of the action and interaction of natural and/or human factors" [4].

The rural landscape differs from other types of landscape in that it occupies a territory where rural, agricultural and livestock farming, and forestry activities take place [5]. The presence of humans is very important, on the one hand because of their constant action on the landscape, and on the other, because of landscape's role as a source of enjoyment. The rural landscape comprises views, types of constructions, vegetation, sounds, the uses and customs of the population, gastronomy, and many other aspects [6]. These types of landscapes are predominant in Europe and represent around 77% of the area of the European Union [7].

There is no question that landscape constitutes a natural resource, and as such it is basic but intangible. All resources, even intangible resources, have a worth, which can be translated into an economic value that increases when the asset becomes scarce.

For centuries, agricultural activities have exclusively and extensively marked and defined the landscape and imprinted it with its own distinctive features. In the last

decades of the 20th century, agricultural production evolved towards intensive farming clustered in areas that guaranteed the highest yields, leading to the abandonment of areas of lower yield. Until then, the only specifications in the agricultural policies of the time concerned observing the landscape regulations in protected areas of low-yield agricultural land with high ecological diversity and natural interest such as national parks. This had a negative effect on the landscape, which was first addressed in the reform of the Community Agricultural Policy (CAP) in the 1980s. Specific measures were developed for certain unique parts of the territory such as mountain areas, regions at evident risk of depopulation, protected areas with significant agricultural activity such as river deltas, and environments classified as natural parks. It was after the reform of the CAP in 1992 that the environmental damage caused up to that point was finally tackled, and an aid scheme was developed for agricultural practices that were compatible with the preservation of the environment [8]). Member states were allowed to adopt aid programs to sustain the environment in general and the landscape in particular. From this point on, aid or direct payments to the agricultural sector became widespread, due to farmers' role as the agents directly responsible for generating and maintaining the agricultural landscape. Today [9], this aid is maintained in programs such as the so-called green payments –or greening– for farmers who, according to the structure of their agricultural operation, follow certain environmental practices.

The consideration of the rural landscape as a resource has gained traction in recent years. In Europe, the European Landscape Convention (ELC) [3], ratified by Spain on 26 November 2007 through the Instrument of Ratification of the ELC (number 176 of the Council of Europe), was signed in Florence on 20 October 2000 (BOE no. 31 of 5 February 2008). It provides a solid and novel framework that places landscape at the forefront of European policies on cultural heritage, the environment, and territorial ordination. The most important new development, compared to earlier conventions, is the integrated vision of the landscape in its cultural and natural aspects, and the introduction of its social dimension.

Landscape assessment is becoming increasingly important, and in countries undergoing rampant urbanization that displaces and reduces the natural landscape –as in the case of Europe–, concern for the environment and its value is on the rise [10,11]. Sustainable management of the landscape implies the rational use of resources to increase the well-being of the population so that its use can be prolonged over time [12].

Since the 1960s, numerous assessment methods have been developed that can be classified into subjective, individual or group assessment [13,14]; methods that use the physical attributes of the landscapes as a substitute for personal perception [7,15]; and methods that opt for the combined use of physical and psychological attributes in their evaluation [16]. Although there is no single method to assess landscapes, the methodological tendency inclines towards mixed methods, as they avoid the systematization that reduces the subjectivity of the aesthetic evaluation process [17].

The various methods for assessing landscape seek to find a solution to the problem of quantifying opinions on the intrinsic quality of the landscape and on people's aesthetic response to it [18].

Direct methods of landscape assessment are the most widespread and are based on an evaluation of the observer's individual preference for the landscape as a whole. They use verbal surveys or questionnaires and apply quantitative or ordinal scales, where the number is equivalent to the order of preference for a landscape, generally represented in a photograph. These are used when the aim is to determine the public's appreciation of the landscape by assessing a sample that must be representative. Collecting this information in situ has its drawbacks, namely the costs of execution and possible difficulties of access by the participants, which in some cases can be solved by using the Internet [19].

There are numerous recent examples of case studies of landscape assessment based on observers' preferences at the European level, for natural [20,21], agricultural [22], and urban landscapes [23,24]. The results of this type of analysis have identified strategic lines

to enhance the landscape function of agricultural systems with a view to their sustainable development, particularly in the sphere of rural tourism [25–28].

Although a number of studies have highlighted the influence of different demographic factors on landscape assessment [16,29,30], it has yet to be established conclusively which demographic characteristics have the greatest effect on preferences, and which landscape attributes determine the preference of each group. The most recent works reveal that educational level and gender are significant factors in assessing preferences [31].

The aim of this work is to analyse the influence of different demographic factors on the preferences for rural landscapes in the locality of Campo de Criptana (Ciudad Real). Specifically, an analysis has been made of observers' preferences in relation to their educational level (university educated or not), place of origin (whether they come from the locality itself or from outside), gender, and age.

## 2. Materials and Methods

### 2.1. Study Area

The municipality of Campo de Criptana, with an approximate population of 15,000 inhabitants, is located in the central part of the La Mancha plain between coordinates 2°51′10″7 W and 3°11′10″7 W (longitude) and 39°20′04″7 N and 39°30′04″7 N (latitude) (Figure 1). The topographical characteristics of the area are such that the road layout allows rapid connections between population centres and any point of the municipal area of Campo de Criptana. The average density of roads is 2.6 m/ha. The most important road is the N-420 from Cordoba to Tarragona and the local roads CR-110 from Alcázar de San Juan to Miguel Esteban. The Madrid-Alicante railway line also runs through the municipal territory, parallel to the N-420.

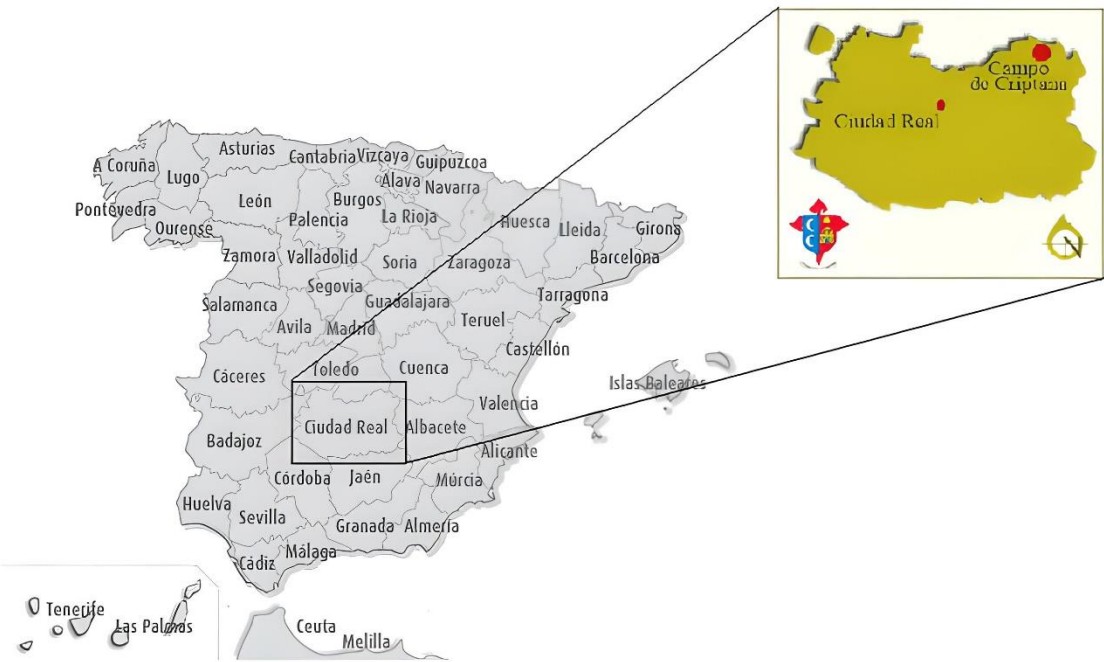

**Figure 1.** Location of the study area.

The municipal district covers an area of 21,231 hectares; the altitude ranges between 650 and 750 m, most of which is occupied by a broad plain that is part of a physiographic domain with several important lake complexes and crossed by the river Záncara in the southern part, a tributary of the Guadiana fed by the river's aquifer 23 [32]. The northern third of the territory is somewhat elevated and is the site of the well-known Sierra de Los Molinos, one of the most important cultural resources in the area and home to the famous

windmills mentioned in the novel "Don Quijote" by Miguel de Cervantes. It was declared a Property of Cultural Interest (BIC) in 2002 with the category of Historic Site [33].

The predominant economic activities in the municipality that affect the landscape are construction and agriculture. Construction energises the local economy and employs a large part of the population, in addition to acting as an engine that drives professional sectors such as carpentry, electricity and others. With regard to agriculture, the main crops cultivated in the area, and which largely conform the local landscape, are vines, olives and cereals. Sheep are the predominant livestock in the area.

Other factors that determine the landscape of Campo de Criptana are the temperature, which ranges between an average of 13 °C and 17 °C. The average temperature of the coldest month is 5–6 °C and the average temperature of the warmest month is 26 °C. The period of frosts lasts between five and six months. Annual ETP is 915 and the duration of the dry period is 4–5 months. Mean annual precipitation is 400 mm. The balance between the mean precipitation in the area and the potential water needs of the vegetation therefore place it in a dry Mediterranean humidity regime.

The main tree species are *Quercus ilex*, which is found in areas of cereal and grazing, and *Pinus halepensis* which grows in reforested areas.

The fauna present in the study area are either species for rural domestic use such as dogs, sheep, donkeys and others, and wildlife including important small game species like partridge, rabbit, and hare.

The selected territory is a representative rural land of the singular landscape of La Mancha plain. The La Mancha plain is an outstanding landscape unit due to its topographic, climatic, hydrographic, demographic, and cultural characteristics. All these aspects give it differentiating characters ascribed to the geographical area of the region of La Mancha [34].

### 2.2. Scene Selection

The rural landscape of Campo de Criptana is characterised by the frequency of four landscape attributes that previous investigations [7] considered important in the assessment of its visual quality: presence of fauna, cultural resources, presence of vegetation, and breadth of the viewshed.

To assess the visual preference of observers of the rural landscape on the La Mancha plain, the scenes presented for assessment had to contain all the possible combinations of these elements. It was therefore necessary to find specific locations and situations that contained this combination of elements. Photographs were used instead of real landscapes to make it easier for the same individual to assess the same scenes and is a technique that has frequently been used in previous works [16,21,31,35].

The photographs were taken in June 2000, between 10 a.m. and 4 p.m., on clear or slightly overcast days to control the lighting conditions. The person taking the photographs made sure they depicted a wide variety of the landscapes in the location, including different combinations of the four attributes considered. The equipment used was a Casio Exilum-ZX100 digital camera. A total of 110 photographs were taken, which were classified into 14 strata. Each stratum comprised photographs showing scenes with the same combination of the four landscape attributes. Photographs were obtained of all the combinations, except the combination "presence of fauna and vegetation with absence of the other elements" and "presence of fauna with absence of the other elements".

A panel of qualified landscape experts selected one photograph from each stratum. The Delphi method was used for its suitability for complex studies that require subjective judgments and decisions [36]. Four researchers and landscape experts from the School of Forestry at the Universidad Politécnica de Madrid selected the 16 photographs representing the landscape in the study area. This selection was carried out in two rounds: in the first, each expert selected a maximum of three photographs for each category, before reaching a consensus in the second round. The catalogue of photographs was completed with two images showing the combination of attributes that could not be obtained in the field, which were available under license on a Spanish nature photography network in 2000 [37].

These scenes are described in Table 1 with their order number, generated at random for their presentation to the observers (see Supplementary Materials):

**Table 1.** Catalogue of photographs obtained with all the combinations of the four attributes of the landscape.

| Photo | Description | Fauna | Vegetation | Cultural Resources | Breadth of the Viewshed |
|:---:|:---|:---:|:---:|:---:|:---:|
| 1 | A small fox can be seen in the photo with its hind legs obscured by a clump of grass [36] | presence | presence | absence | absence |
| 2 | A broad view of the countryside in La Mancha with various cultivated fields in the distance. | absence | absence | absence | presence |
| 3 | A broad view of the countryside with vines and an almond tree | absence | presence | absence | presence |
| 4 | Only the five windmills | absence | absence | presence | absence |
| 5 | A broad view with a donkey in the foreground, some vegetation and cultural resources at the bottom. | presence | presence | presence | presence |
| 6 | A broad view with a flock of sheep and some vegetation. | presence | presence | absence | presence |
| 7 | Shows only a rabbit [36] | presence | absence | absence | absence |
| 8 | A broad view with almond trees in front and the village at the bottom. | absence | presence | presence | presence |
| 9 | A partridge with vegetation in front, and the shrine of La Virgen de Criptana at the bottom. | presence | presence | presence | absence |
| 10 | The shrine of La Virgen de Criptana surrounded by diverse vegetation. | absence | presence | presence | absence |
| 11 | A broad view with no vegetation and the village in the background | absence | absence | presence | presence |
| 12 | A reforestation of pine trees in the foreground. | absence | presence | absence | absence |
| 13 | A broad view with a donkey in the foreground. | presence | absence | absence | presence |
| 14 | A broad view with a flock of sheep and the windmills in the background. | presence | absence | presence | presence |
| 15 | The church of El Convento, located inside the village, with several pigeons on the roof. | presence | absence | presence | absence |
| 16 | Grape harvesting machine. | absence | absence | absence | absence |

*2.3. Survey of Observers' Preferences*

The photographs were stored in digital format and were also arranged in an album on 18 × 24 cm photographic quality paper. They were stored in protective plastic sleeves in a photo album. These photographs were shown to the volunteers in whichever format they requested (image on a laptop computer or on paper) and in random order so they could make their assessment and assign a score to the photos.

The same survey was given to all the individuals. It began with some brief questions on the individual's characteristics, and an explanation of the aim of the work and the assessment scale.

The landscapes represented were assessed using a Likert-type scale that has been very widely used in previous works [7,31], ranging from 1 to 5 according to the preferences of each individual surveyed, as follows:

1 = I don't like it at all; 2 = I don't like it much; 3 = I quite like it; 4 = I like it; 5 = I like it a lot.

A calculation was made of the minimum number of observers that would be needed to estimate differences in the average scores for a landscape, based on the data obtained in the previous work by Cañas et al. [38]. This work reported an average score of 3.26 and an estimated standard deviation of 1.16, calculated with 183 surveys of different population

groups. It was confirmed that the scores follow a normal distribution. The Statgraphics Centurion program was used to calculate the sample size, setting a power of 95% with a significance level of 5% to detect an 0.5 difference in scores between both populations. The minimum size calculated was 73 individuals per population to obtain an error of less than half a point in estimating the differences in average scores between both populations with a confidence level of 95%.

Figure 2 shows photographs of the two landscapes with the highest average scores and the two with the lowest scores.

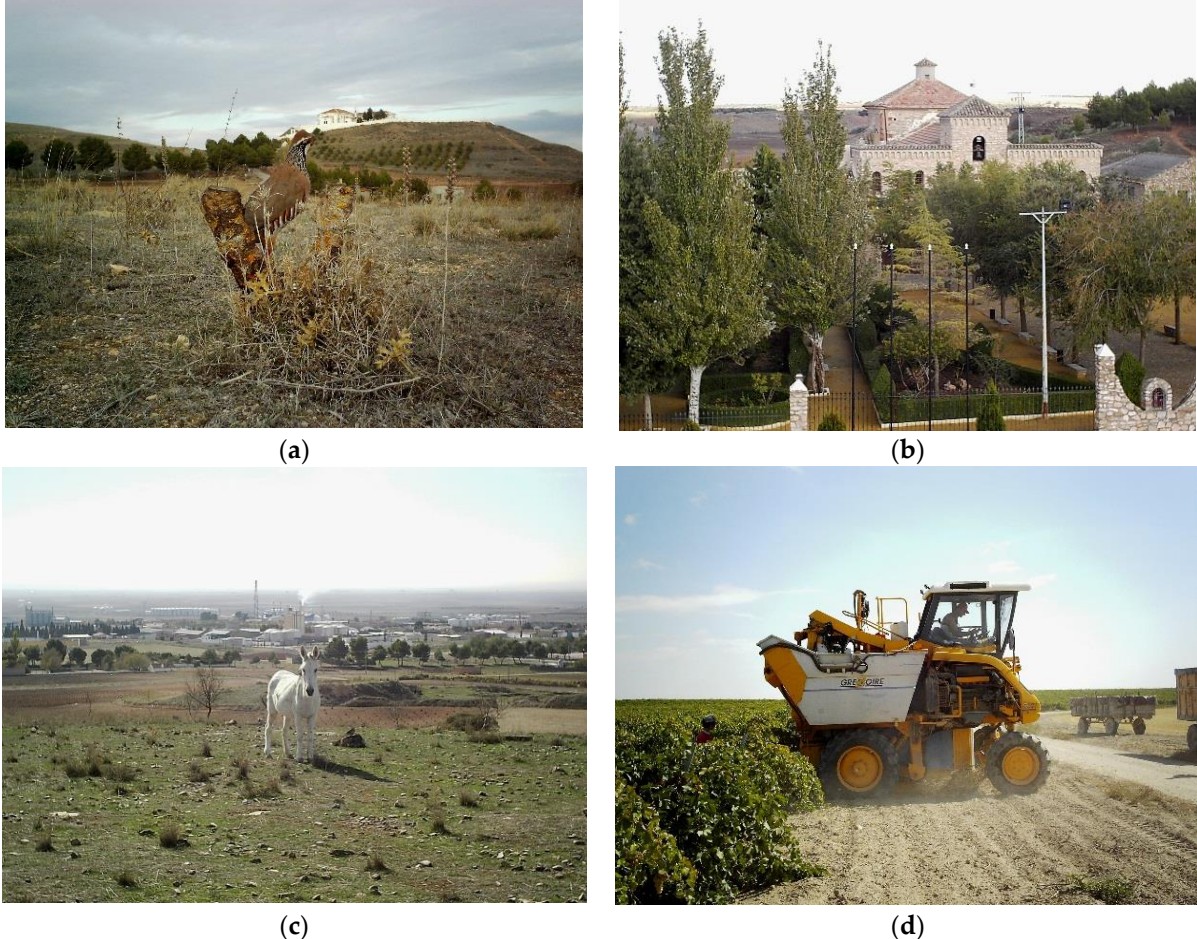

**Figure 2.** Example of landscapes presented for assessment: (**a**) PHOTO 9, score 3.99; (**b**) PHOTO 10, score 3.88; (**c**) PHOTO 5, score 2.96; (**d**) PHOTO 16, score 2.68.

A total of 80 surveys were made of non-university educated individuals and 80 of university educated individuals. Seventy-nine surveys were also made of people living in the location and 82 of people living outside it. These last were more difficult to obtain and were mainly taken in Madrid's Atocha railway station and at the Ciudad Universitaria university campus, also in Madrid, seeking to ensure diversity in the sample composition. It was not possible to maintain a balance in the other characteristics analysed in the observers (Table 2). The respondents were aged between 18 and 60 years. The two age categories— under 40 and over 40—were determined based on the results of another work [39] featuring two age groups, from 18 to 39 years and 40 to 60 years, with similar responses within each group and differences between the groups. Local respondents were all permanent residents in the landscape. The survey was conducted through a structured personal interview, showing the photographs and not the landscape itself, in which the interviewers requested the respondent's assessment of each photo. All respondents completed the survey in an average time of 25 min.

**Table 2.** Comparison of the demographic characteristics of the study participants.

| Demographic Characteristic | Variable | Number | Percentage |
|---|---|---|---|
| Educational level | University educated | 80 | 50% |
| | Non-university educated | 80 | 50% |
| Place of residence | In Campo de Criptana | 78 | 49% |
| | Outside Campo de Criptana | 82 | 51% |
| Gender | Female | 68 | 42.5% |
| | Male | 92 | 57.5% |
| Age | Under 40 | 116 | 72.5% |
| | Over 40 | 44 | 27.5% |

### 2.4. Data Analysis

To study their quantitative influences on landscape preference, the demographic characteristics were codified as fictitious variables (for educational level: university educated = 1, non-university educated = 0; for place of residence: in Campo de Criptana = 0, outside Campo de Criptana = 1; for gender: male = 0, female = 1; age: under 40 = 0, over 40 = 1).

The analysis was done with Statgraphics Centurion XVII software (2014, Statpoint Technologies, Inc., The Plains, VA, USA). If we assume that the scores for the set of photos follow a normal distribution, parametric contrasts can be applied to compare the values of their means and standard deviations (parameters that characterize any normal distribution). Contrasts of normality were done with the Shapiro-Wilks test [40]. The contrasts used are the multifactor ANOVA test with interactions of up to three variables, the F test for equality of variance, and the *t* test for equality of means, habitual in this type of work [7,30]. As they are ordinal variables, the scores for each particular photograph are analysed by means of non-parametric contrasts: Mann-Whitney, difference between means, and the $\chi 2$ test, also used in other works on the assessment of preferences [41]. A significance level of 5% was applied in all the tests.

## 3. Results

This section may be divided by subheadings. It provides a concise and precise description of the experimental results, their interpretation, as well as the experimental conclusions that can be drawn.

### 3.1. Study of the Mean Score of the Set of 16 Landscapes

Table 3 shows a summary of the information obtained for the populations in the sample, with the values of the means (M), medians (Md), standard deviation (SD) and coefficient of variance (VC).

In all the demographic characteristics, the means and medians of the variables are almost the same (differences of less than 2.5%). The statistics between the populations in the sample are also similar.

The hypothesis that the data are from variables with a normal distribution is not rejected except in the case of the over 40 s, where the normal distribution model is rejected (*p*-value of 0.0298) with 95% confidence. As this is the only case, parametric tests are applied to all the demographic characteristics, suitable precautions may be taken in the case of decisions for the hypotheses on age.

**Table 3.** Statistical summary of the mean scores by population.

| Demographic Characteristic | Variable | M | Md | SD | VC (%) |
|---|---|---|---|---|---|
| Educational level | University educated | 3.295 | 3.250 | 0.490 | 14.86 |
| | Non-university educated | 3.484 | 3.500 | 0.558 | 16.02 |
| Place of residence | In Campo de Criptana | 3.315 | 3.313 | 0.582 | 17.56 |
| | Outside Campo de Criptana | 3.430 | 3.438 | 0.463 | 13.49 |
| Gender | Female | 3.463 | 3.500 | 0.569 | 16.44 |
| | Male | 3.291 | 3.219 | 0.463 | 14.06 |
| Age | Under 40 | 3.322 | 3.313 | 0.511 | 15.37 |
| | Over 40 | 3.568 | 3.563 | 0.551 | 15.44 |

Table 4 shows the results of the F statistic and the significance (*p*) of the ANOVA test to detect differences in mean preferences for the demographic characteristics analysed and their interactions, to level. (Significant values at 95% confidence level highlighted in red).

**Table 4.** Analysis of variance for the assessment.

| Source | Factors | F | Significance |
|---|---|---|---|
| Main effects | A: Educational level | 8.46 | 0.0036 |
| | B: Place of residence | 2.13 | 0.1445 |
| | C: Gender | 5.98 | 0.0145 |
| | D: Age | 10.42 | 0.0012 |
| Interactions | AB | 6.12 | 0.0134 |
| | AC | 7.81 | 0.0052 |
| | AD | 0.59 | 0.4433 |
| | BC | 2.72 | 0.0991 |
| | BD | 46.63 | 0.0000 |
| | CD | 0.75 | 0.3879 |
| | ABC | 5.79 | 0.0161 |
| | ABD | 13.79 | 0.0002 |
| | ACD | 0.88 | 0.3470 |
| | BCD | 0.66 | 0.4167 |

Table 4 shows the influence of educational level, gender, and age on the preference. The interaction between educational level and place of residence and educational level and gender and place of residence and age is also influential. Other interactions include educational level, place of residence and gender, and between educational level, place of residence, and age.

The results of the F test and the *t* test for differences between variables (Table 5) corroborate the results of the previous ANOVA. The F test for equality of variances uses the null hypothesis of equal variances as opposed to the alternative hypothesis of different variances. The *t* test for equality of means uses the null hypothesis of equal means as opposed to the alternative hypothesis of different means. The result of the contrast of variances determines the calculation of the t statistic for the contrast of equality of means. (Significant values at 95% confidence level highlighted in red).

Significant differences between the variances can only be detected for place of residence at a 95% confidence level. This is also the only characteristic that does not allow the rejection of the difference between the observer's visual preferences (Table 5).

The significances of the contrasts in the *t* tests are slightly higher than those obtained with the ANOVA analysis for main effects (Table 4), although the results are the same.

**Table 5.** Summary of the contrasts to compare preferences.

| Test | Factors | Significance |
|---|---|---|
| Equality of variances | A: Educational level | 0.2449 |
| | B: Place of residence | 0.0439 |
| | C: Gender | 0.0756 |
| | D: Age | 0.5202 |
| Equality of means | A: Educational level | 0.0241 |
| | B: Place of residence | 0.1657 |
| | C: Gender | 0.0437 |
| | D: Age | 0.0086 |

*3.2. Study of the Assessment of the 16 Landscapes*

Due to the discrete nature of the variable (scores from 1 to 5), the assessments of each photo do not follow normal distributions. However, the hypothesis is verified with the usual tests, and the hypothesis of normality is rejected for the assessments of the 16 photos.

The average scores between groups of variables for each photo range between minimums of 2.12 and maximums of 4.63.

To contrast the hypothesis of equality of opinions between the two populations, the medians (M-W) were compared by means of non-parametric contrast. The null hypothesis of the contrasts in each photo consists of the equality of medians between both populations; the alternative hypothesis would be the one contrary to the null hypothesis. The $\chi2$ test for homogeneity of populations is also incorporated to detect differences between the frequency of the scores awarded each photo by the population group. The null hypothesis is the equality of the proportions between the scores awarded by each group to the photo, and the alternative hypothesis is the opposite. Table 6 shows the results (*p*-value) of the contrasts, with significant values at 95% confidence level highlighted in red.

**Table 6.** Significance (*p*-value) in non-parametric tests for comparison of preferences.

| Demographic Characteristic | Photo | M-W | χ2 |
|---|---|---|---|
| Educational level | 1 | 0.0504 | 0.2468 |
| | 2 | 0.1259 | 0.1058 |
| | 3 | 0.9433 | 0.1653 |
| | 4 | 0.2332 | 0.2874 |
| | 5 | 0.0025 | 0.0414 |
| | 6 | 0.2457 | 0.6616 |
| | 7 | 0.0058 | 0.0117 |
| | 8 | 0.0215 | 0.1083 |
| | 9 | 0.0866 | 0.0398 |
| | 10 | 0.0207 | 0.1805 |
| | 11 | 0.0311 | 0.0312 |
| | 12 | 0.9666 | 0.2023 |
| | 13 | 0.6786 | 0.6337 |
| | 14 | 0.0285 | 0.0587 |
| | 15 | 0.1276 | 0.3293 |
| | 16 | 0.1884 | 0.5363 |
| Place of residence | 1 | 0.0996 | 0.0884 |
| | 2 | 0.0072 | 0.0596 |
| | 3 | 0.3691 | 0.524 |
| | 4 | 0.6306 | 0.3348 |
| | 5 | 0.1787 | 0.0652 |
| | 6 | 0.7046 | 0.4125 |
| | 7 | 0.4074 | 0.7607 |
| | 8 | 0.0048 | 0.0366 |
| | 9 | 0.0109 | 0.1024 |
| | 10 | 0.2862 | 0.0515 |

**Table 6.** *Cont.*

| Demographic Characteristic | Photo | M-W | χ2 |
|---|---|---|---|
| | 11 | 0.0370 | 0.0186 |
| | 12 | 0.3851 | 0.6558 |
| | 13 | 0.7513 | 0.3962 |
| | 14 | 0.9685 | 0.133 |
| | 15 | 0.8329 | 0.3884 |
| | 16 | 0.0510 | 0.1222 |
| Gender | 1 | 0.0873 | 0.1044 |
| | 2 | 0.4797 | 0.0555 |
| | 3 | 0.9885 | 0.2856 |
| | 4 | 0.9114 | 0.6931 |
| | 5 | 0.5731 | 0.3878 |
| | 6 | 0.2624 | 0.4385 |
| | 7 | 0.0142 | 0.0559 |
| | 8 | 0.3857 | 0.7929 |
| | 9 | 0.1249 | 0.199 |
| | 10 | 0.0152 | 0.1733 |
| | 11 | 0.2243 | 0.4225 |
| | 12 | 0.1382 | 0.51 |
| | 13 | 0.7953 | 0.5379 |
| | 14 | 0.6002 | 0.8267 |
| | 15 | 0.0063 | 0.1033 |
| | 16 | 0.0055 | 0.0012 |
| Age | 1 | 0.6720 | 0.0212 |
| | 2 | 0.5148 | 0.5797 |
| | 3 | 0.4363 | 0.2001 |
| | 4 | 0.9667 | 0.9154 |
| | 5 | 0.0235 | 0.1281 |
| | 6 | 0.0091 | 0.001 |
| | 7 | 0.1928 | 0.5176 |
| | 8 | 0.0089 | 0.0678 |
| | 9 | 0.5024 | 0.8424 |
| | 10 | 0.2004 | 0.64 |
| | 11 | 0.0073 | 0.0042 |
| | 12 | 0.1433 | 0.2735 |
| | 13 | 0.1361 | 0.4246 |
| | 14 | 0.0003 | 0.0025 |
| | 15 | 0.0276 | 0.1765 |
| | 16 | 0.3637 | 0.6205 |

Table 6 shows the differences in the results obtained with both tests. The M-W test of difference between medians detects a greater number of significant differences than the χ2 test.

In the case of educational level and age, the M-W test detects differences in six of the photos, and four for place of residence and gender. Significant differences were found with the χ2 test between four photos in the case of educational level and age, in only two photos for place of residence, and in one for gender.

The photos in which both tests revealed significant differences between populations were Photo 5 (Figure 2) for educational level, Photo 6 (Figure 3) for age, Photo 7 (Figure 3) also for educational level, Photo 8 (Figure 3) for place of residence, Photo 14 (Figure 3) for age, and Photo 16 (Figure 2) for gender. Table 7 shows the statistical summary of the differences between these photos.

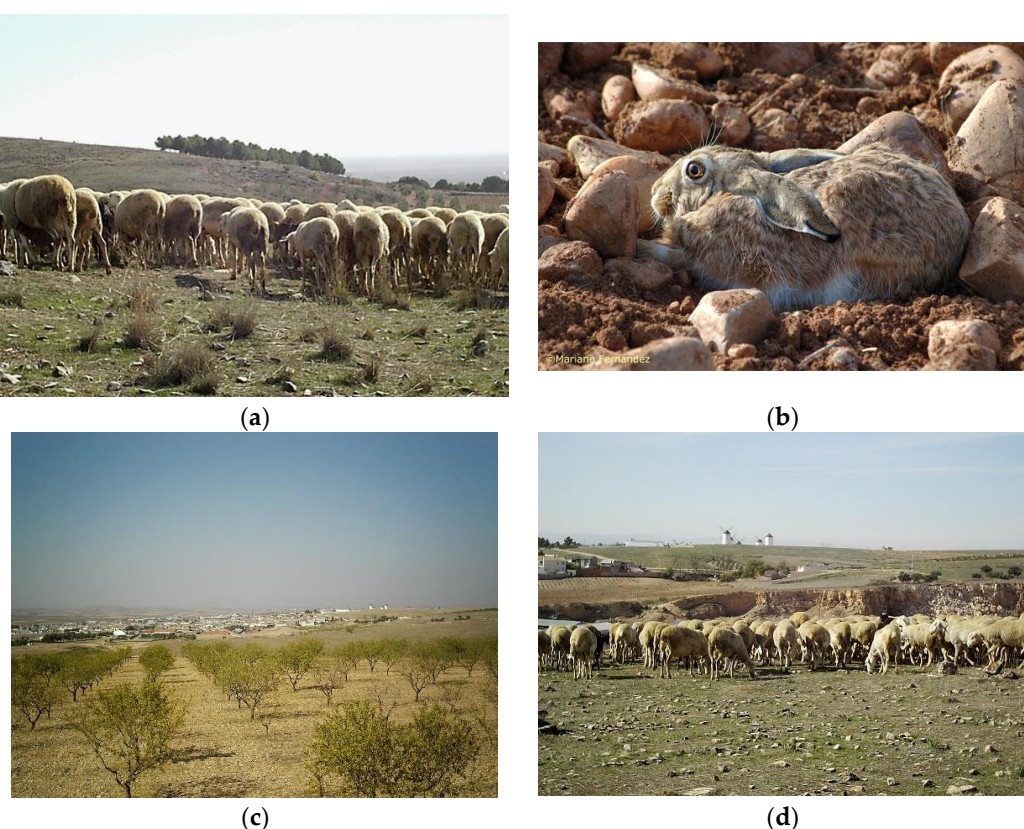

**Figure 3.** Example of the landscapes presented for assessment: (**a**) PHOTO 6, differences by age; (**b**) PHOTO 7, differences by educational level; (**c**) PHOTO 8, differences by place of residence; (**d**) PHOTO 14, differences by age.

**Table 7.** Statistical summary of the mean scores by population, (Photos 5–8, 14 and 16).

| Photo | Demographic Characteristic | Variable | M | Md | SD | VC (%) |
|---|---|---|---|---|---|---|
| 5 | Educational level | University educated | 2.675 | 3 | 1.145 | 42.8 |
| | | Non-university educated | 3.238 | 3 | 1.172 | 36.20 |
| 7 | Educational level | University educated | 3.363 | 3 | 0.917 | 27.28 |
| | | Non-university educated | 3.788 | 4 | 1.064 | 28.01 |
| 8 | Place of residence | In Campo de Criptana | 3.808 | 4 | 0.927 | 24.34 |
| | | Outside Campo de Criptana | 3.410 | 3 | 0.904 | 26.50 |
| 16 | Gender | Female | 2.870 | 3 | 1.160 | 40.42 |
| | | Male | 2.412 | 2 | 0.950 | 39.39 |
| 6 | Age | Under 40 | 3.138 | 3 | 0.790 | 25.18 |
| | | Over 40 | 3.523 | 3.5 | 0.976 | 27.71 |
| 14 | Age | Under 40 | 3.138 | 3 | 0.903 | 28.78 |
| | | Over 40 | 3.773 | 4 | 0.937 | 24.83 |

Photo 11 (Figure 4) shows the differences for all the characteristics except gender. These differences are shown in Table 8.

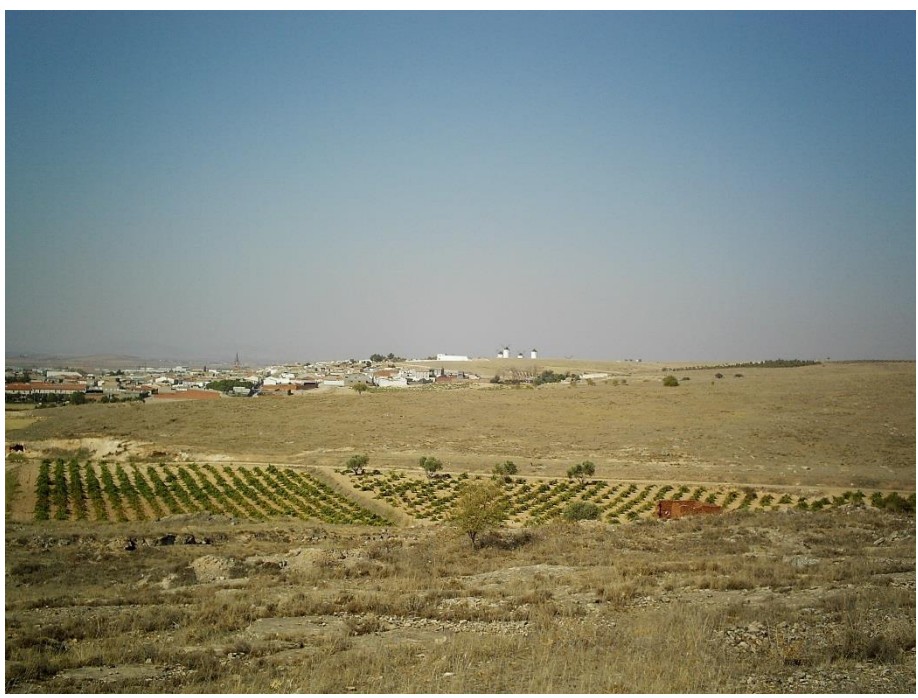

**Figure 4.** PHOTO 11. A broad view without vegetation with the village in the background, also showing windmills.

**Table 8.** Statistical summary of the mean scores by population (Photo 11).

| Demographic Characteristic | Variable | M | Md | SD | VC (%) |
|---|---|---|---|---|---|
| Educational level | University educated | 2.950 | 3 | 0.913 | 30.94 |
| | Non-university educated | 3.313 | 3 | 1.074 | 32.43 |
| Place of residence | In Campo de Criptana | 3.244 | 3 | 0.996 | 30.70 |
| | Outside Campo de Criptana | 2.949 | 3 | 1.068 | 36.21 |
| Age | Under 40 | 2.983 | 3 | 0.960 | 32.18 |
| | Over 40 | 3.523 | 3 | 1.045 | 29.67 |

## 4. Discussion and Conclusions

There is a close similarity between the average scores awarded to the various landscapes by all the population groups in the study (see Table 2), in line with latest works [41].

In line with the works reviewed [31,42,43], the characteristic of educational level determines the observers' preference. Non-university educated respondents show higher average scores for this rural landscape. The most significant differences were detected in broad views of the landscape with the presence of fauna but without vegetation. These findings are similar to the results of another study [31] in which the participants with the highest education level preferred natural vegetation, which is closely aligned with ecological concerns.

This work analyses the influence of the demographic characteristic of belonging to the territory under assessment (place of residence). This is one of the few works to analyse this characteristic. Although the authors expected the local respondents to award higher scores to the landscape, the results of the statistical analysis do not justify the acceptance of this supposition. These findings are similar to another work [42]. However, significant differences were detected in the variability of the scores between both groups and were higher among the local residents (see Table 5).

This work also reveals significant differences in preferences according to gender and age. Men value the rural landscapes a little higher, as do people over 40. This agrees with previous works that analyse demographic factors in preferences [16,31].

Previous works do not agree on the effects of demographic variables on landscape preference, among other reasons due to the interactions existing between demographic factors [43]. Other authors consider that these interactions vary between different countries and regions [31], although very few studies have been carried out on Mediterranean landscapes [41]. This was assessed by analysing the possible interactions between the demographic factors, revealing five combinations with significant differences: educational level with place of residence and gender; place of residence with age (for two-factor interactions); and study level with place of residence, combined with gender and age. Non-university educated observers living locally assigned the highest scores to the landscapes; non-university educated male respondents and local residents aged over 40 rated the landscapes highest and with very similar scores (between 3.52 and 3.64). Non-university educated observers living locally, both males and respondents aged over 40, gave the highest scores to the rural landscape. These results may be due to the greater connection between older local inhabitants—who also coincide with the non-university-educated population—with a rural landscape made up of agricultural and livestock activities.

There are differences in seven of the 16 photos analysed. By educational level there are differences in photos 5, 7 and 11, with no similarities found between them. Non-university educated respondents gave a higher score to Photo 5 showing fauna, vegetation, and natural resources. This photo shows the scene with the four landscape attributes that are considered important in the assessment of visual quality of this landscape.

Photos 7 and 11 are most highly rated by university-educated observers. Photo 7 shows only a rabbit, and Photo 11 shows a broad view without vegetation with the village in the background, and some windmills. The preference for Photo 7 may be because people with a university education have a more positive attitude toward wildlife, while in the case of Photo 11 because distinctive constructions (windmills) are assessed positively, as concluded by similar studies [41,44].

Photos 6, 11, and 14, showing broad views over the countryside of La Mancha, are detected by age. Observers aged over 40 rate all these scenes higher, which may be due to older people's greater affinity to traditional territorial uses [41].

By place of residence, differences can be detected between photos 8 and 11 showing scenes with panoramic views and the village in the background. As assumed by other works [20], these groups have a different experience of the local landscape, so local respondents rate both these scenes more highly.

Gender differences are highlighted in Photo 16, showing a grape-harvesting machine and no landscape elements, and is more highly rated by women. Recent similar works explain these differences due to the fact that women have a more positive attitude towards natural landscapes [41,42,45].

In view of these results, it can be concluded that all the demographic factors analysed have an influence on preferences in rural landscapes. As explained in [31,45], we believe that an adequate inclusion of visual preferences in the management of rural spaces improves the satisfaction of the inhabitants and generates greater support for their management. This is especially relevant in rural tourism planning and landscape sustainability [36].

Educational level and age show clearly significant differences even at a 99% confidence level. Non-university educated observers and respondents aged over 40 show a greater preference for the rural landscape. Although age was not found to be influential in other works [30], the age intervals were different, which suggests that the intervals used in this study can be considered more suitable.

The knowledge of the interactions between demographic factors must be improved, and more rural landscape studies are required in the Mediterranean area. This work confirms the influence of demographic factors in landscape preferences. The complexity of

the interactions between them could explain the differences detected in the various studies in terms of which factors are key.

**Supplementary Materials:** The following are available online at https://www.mdpi.com/article/10.3390/su132413799/s1, Supplementary Materials S1: Photographs used.

**Author Contributions:** Conceptualization, E.A.-T. and D.M.-V.; methodology, E.A.-T.; validation, E.A.-T. and M.Á.G.-O.; formal analysis, E.A.-T. and J.J.R.-M.; investigation, D.M.-V.; writing—original draft preparation, E.A.-T.; writing—review and editing, M.Á.G.-O.; supervision, J.J.R.-M. All authors have read and agreed to the published version of the manuscript.

**Funding:** This research received no external funding.

**Informed Consent Statement:** Not applicable.

**Conflicts of Interest:** The authors declare no conflict of interest.

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
