# Peer review of "Differences in Visual Preference in Rural Landscapes on the Plain of La Mancha in Spain"

_sustainability, doi:10.3390/su132413799_

Round 1

Reviewer 1 Report

This manuscript has potential to contribute to the scholarship on sense of place, and the contribution of belonging to the way in which landscapes are valued. The statistical approach potentially provides a sound framing for evidence in the conclusions with some adjustments and updating of language, approach and technique (methodology).

LOCATION OF SCHOLARSHIP:

My first comment is that the results, discussion and conclusions are not couched within the recent literature, with only 5 out of 38 references within the past five years. The concepts, definitions and approach do not reflect contemporary approaches. The discussion and conclusions do not reference any literature at all, thus not linking the results in this manuscript or making a contribution to a larger body of scholarship.

DEFINITIONS AND RESEARCH QUESTIONS:

The rationale for this research is not presented clearly; the research questions are not succinct or clear within the introduction; the key research question is found on line 358 indicating the theoretical context of the research; there is no definition or discussion on the key theoretical context of the research: belonging.

METHODOLOGY AND METHODS:

The testing of hypotheses in this way is somewhat dated, and limited in the provision of insight into relationships between sense of place and distinct landscape values, research questions are more useful; the rationale and limitations for the analysis techniques are missing; the sampled population descriptions are not clear and the final numbers of people surveyed are not outlined; operationalisation of theoretical concept to survey questions is absent, as are the survey questions; description of triangulation, replication and index development in the survey questions is absent; categorisation (natural elements; agricultural elements; cultural elements) and testing for latent structure in the landscape values represented in the photographs is missing; fauna must be differentiated between domestic animals and wildlife; age breakdown should represent standard demographic analysis categories, not just over 40 and under 40 (which were not justified in any case); cultural, agricultural and natural landscapes should be differentiated more clearly in the variables; key concepts for sense of place, including length of time resident in a landscape as predictor of the way in which landscapes are valued and development of belonging evolves are missing in the variables; description of survey methods/ tools must include how many surveys were conducted, how many were refused, how long did the survey take to answer, how many respondents completed the survey within the landscape in question, and how many respondents completed the survey outside of the landscape in question.

RESULTS

It is important that more information is provided for each table of results, particularly the statistically significant scores (in red) are not defined on the first table but much later. In addition, each written paragraph of results must provide a sentence saying how this relates to the research questions...what do the results mean in normal language. The analysis methods were not justified, nor their limitations discussed; definitions for categories were not provided i.e. line 230 what does 'highest average scores' represent/ mean.

DISCUSSION AND CONCLUSIONS

It is conventional to re-state the research questions at the start of the discussion section, then link results to research questions and the broader literature to indicate new knowledge or correlations in results. Definitions (such as "belonging to territory" line 358) should be provided in introduction; the discussion should not re-present results.

MINOR DETAILS

Italicise Quercus ilex; Gender is generally categorised as female and male rather than woman and man.

Author Response

1- About point LOCATION OF SCHOLARSHIP: The results, discussion and conclusion are being couched within the recent literature.

2- About point DEFINITIONS AND RESEARCH QUESTIONS: Attempts have been made to clarify the concepts.

3- About point METHODOLOGY AND METHODS: Attempts have been made to clarify the concepts in Table 1. Age breakdown has been justified.

4- About point RESULTS: The statistical analysis applied is a classic ANOVA and test of difference of means, and homogeneity of the chi-square. This analysis is widely known; therefore, it has not been considered necessary to include its justification or limitations.

5- About point DISCUSSION AND CONCLUSIONS: The discussion and conclusion are being couched within the recent literature.

6- About point MINOR DETAILS: these errors are being corrected.

Reviewer 2 Report

The research article explores visual preferences in rural landscapes of La Mancha plains, Spain. The paper is sufficiently referenced, the Introduction could be enriched with a few words on the subjectivity of visual assessment (with regard to the method used). 

In line 175 the authors indicate, that "a panel of qualified landscape experts selected one photograph from each stratum." It is worth being more specific here, providing the number of experts, their competencies (landscapes architects? faculty members?) and, more importantly: did you seek their agreement on the selection (Delphi approach)? Or each one could choose a number of images?

I would suggest providing an annexe with the full catalogue of all images used in the survey.

In line 160 the authors mention four landscape attributes considered, among them "breadth of the viewshed". Some images, however, are prevailed by one element (e.g., a rabbit in picture No 7 or grape harvester in picture No 16). Maybe a few words of explanation regarding the criteria of images selection would be helpful.

The part of the discussion should show the research results against the background of the existing knowledge. I think this is missing. It would be valuable to compare your results with those obtained by other researchers or even with your previous achievements. Did the trends in landscape perception change across the last decades? Do your results confirm general trends on visual preferences observed by other authors? Or maybe you observed something contradictory?

Sincerely yours,

Author Response

1- About coments of the first paragraph (introduction): Ok, done it.

2- About coments of the second paragraph (materials and methods): The Delphi method was used. The method justification and details of the procedure are included.

3- About coments of the third paragraph (full cataloge of images): Ok, done it.

4- About coments of the fourth paragraph (criteria of images selection):  Attempts have been made to clarify the concepts in a new Table 1.

5- About coments of the fifth paragraph (discussion): The discussion and conclusion are being couched within the recent literature.

Reviewer 3 Report

Interesting topic. It can be stronger if a link is made with the bigger picture. Why is this paper relevant to Sustainability ? 
